# ConvRot: Rotation-Based Plug-and-Play 4-bit Quantization for Diffusion Transformers

## Abstract

Diffusion models have demonstrated strong capabilities in generating high-quality images. However, as model size increases, the growing memory footprint and inference latency pose significant challenges for practical deployment. Recent studies in large language models (LLMs) show that rotation-based techniques can smooth outliers and enable 4-bit quantization, but these approaches often incur substantial overhead and struggle with row-wise outliers in diffusion transformers. To address these challenges, we develop a theoretical framework: we define column discrepancy to quantify imbalance in Hadamard matrices, prove that regular Hadamard matrices attain minimal discrepancy, and provide a Kronecker-based construction for powers-of-four orders, effectively controlling row- and column-wise outliers. Based on this, we propose ConvRot, a group-wise rotation-based quantization that reduces computation from quadratic to linear complexity while smoothing outliers, and ConvLinear4bit, a plug-and-play module fusing rotation, quantization, GEMM, and dequantization for W4A4 inference without retraining. Experiments on FLUX.1-dev achieve a 2.26× speedup and 4.05× memory reduction, while preserving image quality.

## 1 Introduction

Diffusion models are powerful generative frameworks that produce high-fidelity images (Ho et al., 2020; Rombach et al., 2022), but scaling their architectures significantly increases memory and inference costs. The recently released Qwen-Image model (Wu et al., 2025) reaches a scale of 20B parameters, requiring more than 60 GiB of GPU memory for inference. In large language models (LLMs), quantization has been widely used to compress model size and improve inference speed (Zhu et al., 2024; Dettmers et al., 2022; Xiao et al., 2023), mainly by reducing memory movement and leveraging low-precision compute units in modern GPUs. This makes quantization a promising direction for reducing the memory and latency cost of diffusion models as well. A major source of accuracy loss in quantization comes from outliers, which can distort the scaling factors and degrade performance. Recent studies in LLMs show that rotation-based quantization methods redistribute outliers across channels, enabling 4-bit quantization with minimal accuracy loss (Tseng et al., 2024; Ashkboos et al., 2024; Liu et al., 2024), but the extra rotation operations bring non-negligible overhead that offsets part of the speedup. Therefore, the key challenge is to apply rotation-based quantization to diffusion transformers in a way that preserves accuracy while reducing the rotation cost.

As illustrated in Figure 1, rotation-based quantization methods suppress outliers by applying rotational transformations to both weights and activations. While these methods have been extensively studied in LLMs, directly applying them to diffusion models faces two major challenges. First, the rotation operations themselves introduce substantial computational overhead. Prior works attempt to mitigate this using techniques such as the Fast Walsh-Hadamard Transform (FWHT) and operation fusion to reduce online rotations (Ashkboos et al., 2024). However, the Adaptive LayerNorm (AdaLN) design in Diffusion Transformers (DiTs) (Peebles & Xie, 2023) can break these fusion strategies, forcing more frequent online rotations and offsetting the acceleration benefits. Sparse rotation matrices have

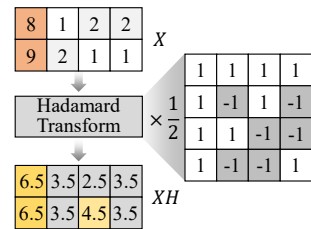

Figure 1: Rotation-based quantization methods can effectively suppress outliers by redistributing energy across channels.

also been explored (Lin et al., 2024a), but without efficient low-level support, they do not lead to practical speedup. Second, as shown in Figure 3, we observe row-wise outliers in certain layers of FLUX.1-dev (Black-Forest-Labs, 2024), which differs from the column-wise outlier patterns commonly found in LLMs. Existing rotation matrix designs, particularly Hadamard matrix-based methods, primarily target column-wise outliers and are therefore ineffective at handling row-wise outliers, leading to noticeable accuracy degradation.

In this work, we propose ConvRot, a novel rotation-based quantization paradigm. First, we introduce a group-wise rotation quantization scheme that reduces computational complexity from quadratic to linear complexity and allows flexible trade-offs between computation cost and outlier suppression by adjusting the group size $N_0$. Second, we adopt the regular Hadamard Transform (RHT) to simultaneously suppress row-wise and column-wise outliers. To support this, we propose a theoretical framework: we formalize the column sum squared property of Hadamard matrices, define the column discrepancy to quantify imbalance, and present a Kronecker-based construction of regular Hadamard matrices for orders that are powers of four, guaranteeing minimal column discrepancy. Based on these regular Hadamard matrices, we implement group-wise RHT with a conv-like matmul operation on weights and activations, which we name ConvRot. Finally, we design ConvLinear4bit, a plug-and-play module that fuses rotation, quantization, GEMM, and dequantization, avoiding expensive loops or extra memory movement while leveraging mature matrix multiplication pipelines on modern GPUs, without requiring complex operator design or additional inference engines. Experimental results demonstrate that ConvRot largely preserves image quality, reduces the memory footprint of the original BF16 DiT by $4.05\times$, and achieves a $2.26\times$ speedup on an RTX 4090 24GB.

In summary, our main contributions are:

- We provide a theoretical framework for rotation-based quantization: we define the column discrepancy to quantify column sum imbalance, and propose a Kronecker-based construction of regular Hadamard matrices for orders that are powers of four, guaranteeing minimal column discrepancy and therefore particularly effective at mitigating the amplification of row-wise outliers in activations.

- We propose ConvRot, a novel group-wise rotation-based quantization paradigm that leverages regular Hadamard Transform (RHT) to simultaneously smooth row-wise and column-wise outliers, reducing computational complexity from $\mathcal{O}(N^2)$ to $\mathcal{O}(N)$ and significantly lowering latency compared to global rotations.

- We design ConvLinear4bit, a plug-and-play module that fuses rotation, quantization, GEMM, and dequantization, enabling training-free W4A4 inference for all linear layers in diffusion models. Experiments on FLUX.1-dev show that ConvLinear4bit stably suppresses outliers by up to $7\times$, reduces memory usage and achieves inference speedup, while preserving image quality.

## 2 RELATED WORK

### 2.1 QUANTIZATION FOR LLMS

Quantization reduces memory traffic and computation by adopting low-precision formats, while also enabling efficient use of hardware-specific accelerators such as INT4 tensor cores (Frantar et al., 2023; Lin et al., 2024b; Dettmers et al., 2022). However, naive per-tensor or per-channel post-training quantization (PTQ) schemes suffer from outliers that dominate the dynamic range, leading to substantial accuracy degradation. Rotation-based quantization addresses this by applying orthogonal transforms to distribute outliers across channels, producing smoother distributions with fewer extreme values (Tseng et al., 2024). Yet, these rotations introduce quadratic complexity, which offsets the potential acceleration. Prior works mitigate the cost with fast hadamard transforms (Ashkboos et al., 2024), fusion the rotation into adjacent linear layers (Liu et al., 2024), or block-diagonal rotations (Lin et al., 2024a). While effective for LLMs, these designs face challenges in diffusion models, fusion breaks under adaptive normalization layers (Peebles & Xie, 2023), and block-diagonal rotations fail to deliver speedup proportional to their reduced computation. In contrast, our ConvRot employs a lightweight group-wise rotation that reduces complexity to linear while preserving sufficient smoothing, and can be directly applied to diffusion models without architectural changes.

## 2.2 Acceleration of Diffusion Models

Diffusion models (Ho et al., 2020) achieve state-of-the-art performance in image and video generation (Wu et al., 2025; Kong et al., 2024), but their inference speed remains a major limitation for deployment due to the inherently slow and computationally intensive iterative process. Existing acceleration strategies include few-step samplers (Song et al., 2020; Lu et al., 2022a;b), distillation (Salimans & Ho, 2021; Luo et al., 2023; Yin et al., 2024), pruning (Zhao et al., 2024c), and caching (Liu et al., 2025). Recently, quantization has also been explored for diffusion models (Li et al., 2023; Zhao et al., 2024b;a; Li et al., 2024b). However, unlike language models, where latency is often dominated by weight loading, diffusion models are computationally bounded (Li et al., 2024b). As a result, weight-only quantization is insufficient for diffusion models, both weights and activations must be quantized to fully exploit low-precision hardware. However, existing methods either maintain activations in higher precision (Dettmers et al., 2023), preventing the use of low-precision tensor cores, or rely on customized inference engines (Li et al., 2024b), which complicates deployment. By contrast, our method supports end-to-end 4-bit weight-activation quantization, fully exploiting low-precision hardware units. Furthermore, the proposed ConvLinear4bit layer is plug-and-play, requiring no specialized inference engine, and integrates seamlessly with existing quantized operators to deliver both memory reduction and practical speedup.

## 3 Preliminary

### 3.1 Equivalent Transformation Based Quantization Methods

Given an activation vector $\mathbf{x} \in \mathbb{R}^n$, a uniform $b$-bit quantizer is defined as

$$Q(\mathbf{x}) = \text{round}\left(\frac{\mathbf{x}}{s}\right), \tag{1}$$

where $s$ is a scaling factor. Large-magnitude outliers in $\mathbf{x}$ inflate $s$, reducing the effective resolution for most elements and making low-bit quantization challenging.

Transformation-based quantization methods apply an orthogonal or diagonal transformation $T(\cdot)$ to redistribute activation magnitudes before quantization while preserving the computation. For a linear layer $\mathbf{Y} = \mathbf{X}\mathbf{W}^\top$, this invariance is expressed as

$$\mathbf{X}\mathbf{W}^\top = T(\mathbf{X})\,T'(\mathbf{W})^\top, \tag{2}$$

where $T'(\cdot)$ is the corresponding transform on the weight. As illustrated in Figure 2, different choices of $T(\cdot)$ correspond to different ways of redistributing outliers. The core challenge is to design $T(\cdot)$ to **suppress outlier amplitudes with minimal computational overhead**, enabling smaller scaling factors $s$ and higher effective precision in low-bit quantization.

### 3.2 Regular $\mathcal{H}$-Matrices for Rotation

**Definition 3.1** (Hadamard Matrix ($\mathcal{H}$-Matrix)). A Hadamard matrix (abbrev. $\mathcal{H}$-Matrix) $\mathbf{H}_n \in \{\pm 1\}^{n \times n}$ satisfies

$$\mathbf{H}_n \mathbf{H}_n^\top = n\mathbf{I}_n, \tag{3}$$

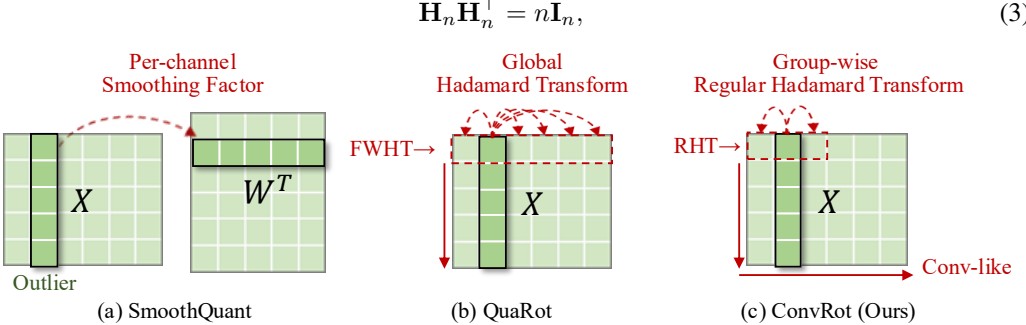

Figure 2: Illustration of how different transformations redistribute outliers. (a) **SmoothQuant** (Xiao et al., 2023): per-channel diagonal transform $T(\mathbf{X}) = \mathbf{X}\,\text{diag}(\mathbf{s})^{-1}$ shifts activation outlier magnitudes into the corresponding channel weights. (b) **QuaRot** (Ashkboos et al., 2024): global Hadamard transform $T(\mathbf{X}) = \mathbf{X}\mathbf{H}$ with orthogonal $\mathbf{H}$ evenly redistributes activation energy. (c) **ConvRot (Ours)**: Group-wise Regular Hadamard Transform performs local smoothing of activations within sliding windows.

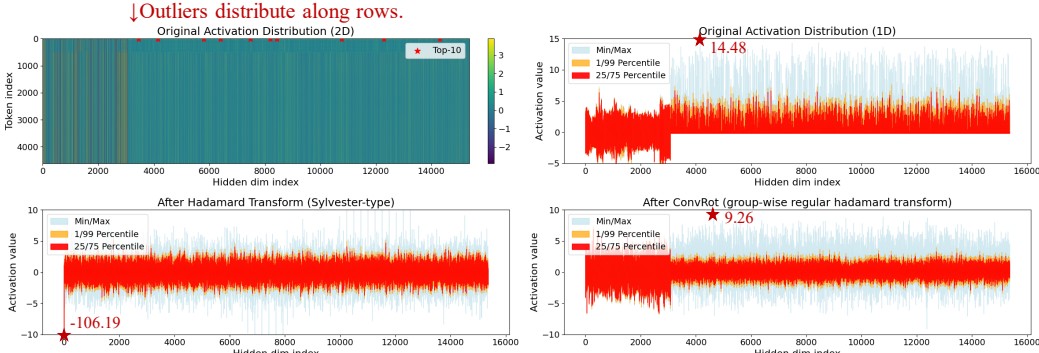

Figure 3: Effect of Hadamard transforms on the `single_transformer_blocks.37.proj_out` activations in Flux. The standard transform amplifies outliers (max = 106.19), while the group-wise regular transform suppresses them (max = 9.26), compared to the original (max = 14.48).

where $\mathbf{I}_n$ is the $n \times n$ identity. Normalized by $1/\sqrt{n}$, $\mathbf{H}_n$ is orthogonal, making it well-suited for rotation-based quantization since orthogonality redistributes outliers. Empirically, $\mathcal{H}$-Matrix rotations outperform random orthogonal ones (Liu et al., 2024; Tseng et al., 2024).

A common construction of Hadamard matrices is the Sylvester-type recursion:

$$\mathbf{H}_1 = [1], \quad \mathbf{H}_{2n} = \begin{bmatrix} \mathbf{H}_n & \mathbf{H}_n \\ \mathbf{H}_n & -\mathbf{H}_n \end{bmatrix}. \tag{4}$$

The Fast Walsh-Hadamard Transform (FWHT) exploits this structure to reduce the matrix-vector multiplication complexity from $\mathcal{O}(n^2)$ to $\mathcal{O}(n \log n)$, using only additions and subtractions. Recent work (Lin et al., 2024a) demonstrates that block-wise rotations can preserve most of the benefits while reducing computational cost. It is worth noting that the FWHT is equivalent to multiplying by a Sylvester-type Hadamard matrix, *whose first column is all ones*, which can inadvertently amplify row-wise outliers in the activations, as illustrated in Figure 3.

**Theorem 3.1** (Column Sum Squared Property). *For an $\mathcal{H}$-Matrix $\mathbf{H}_n$,*

$$\sum_{j=1}^{n} \left( \sum_{i=1}^{n} \mathbf{H}_{ij} \right)^2 = n^2. \tag{5}$$

Since concentrated column sums can amplify row-wise outliers, we quantitatively evaluate the ability of $\mathcal{H}$-Matrices to mitigate this effect by introducing the following metric.

**Definition 3.2** (Column Discrepancy). For an $\mathcal{H}$-Matrix $\mathbf{H}_n$, the *column discrepancy* is defined as

$$\|\mathbf{H}^\top \mathbf{1}\|_\infty = \max_j \left| \sum_i H_{ij} \right|, \tag{6}$$

where $\mathbf{1}$ is the all-ones vector. It measures the largest deviation of a column sum from zero.

This metric is related to *combinatorial discrepancy* (Spencer, 1985; Matousek, 1999), defined for $A \in \{\pm 1\}^{n \times m}$ as

$$\text{disc}(A) = \min_{\varepsilon \in \{\pm 1\}^n} \|A^\top \varepsilon\|_\infty. \tag{7}$$

Here $\varepsilon$ is a $\pm 1$ coloring of rows, chosen to minimize the maximum column imbalance. In our case, the column discrepancy corresponds to the fixed coloring $\varepsilon = \mathbf{1}$, giving a upper bound on $\text{disc}(\mathbf{H})$.

For $\mathcal{H}$-Matrices, the column discrepancy always satisfies

$$\sqrt{n} \leq \|\mathbf{H}^\top \mathbf{1}\|_\infty \leq n. \tag{8}$$

The lower bound follows from the column sum squared property, while the upper bound is achieved by Sylvester-type matrices that contain identical columns. This motivates the study of *regular* $\mathcal{H}$-Matrices, which attain the minimum value.

Figure 4: Overview of ConvRot. Left: ConvLinear4bit serves as a plug-and-play replacement for Linear layers. Right: ConvRot applies Regular Hadamard Transform (RHT) on non-overlapping sliding windows of the activation tensor, with each window multiplied by a regular Hadamard matrix.

**Definition 3.3** (Regular $\mathcal{H}$-Matrix). An $\mathcal{H}$-Matrix is *regular* if each row and column sums to $\pm\sqrt{n}$.

**Theorem 3.2.** *Regular $\mathcal{H}$-Matrices attain the minimal possible column discrepancy:*

$$\max_j \left| \sum_{i=1}^n \mathbf{H}_{ij} \right| = \sqrt{n}. \tag{9}$$

**Theorem 3.3** (Kronecker Construction). *For every $k \geq 1$, a regular $\mathcal{H}$-Matrix of order $n = 4^k$ exists. Starting from*

$$\mathbf{H}_4 = \begin{bmatrix} 1 & 1 & 1 & -1 \\ 1 & 1 & -1 & 1 \\ 1 & -1 & 1 & 1 \\ -1 & 1 & 1 & 1 \end{bmatrix}, \tag{10}$$

*one obtains $\mathbf{H}_{4^{k+1}} = \mathbf{H}_{4^k} \otimes \mathbf{H}_4$ via the Kronecker product. Each $\mathbf{H}_{4^k}$ remains regular.*

All proofs are deferred to Appendix A. We leverage this property to design a *group-wise regular $\mathcal{H}$-Matrix rotation* scheme. It reduces peak activations and latency while preserving the smoothing benefits of $\mathcal{H}$-Matrix rotations, making it practical for large diffusion models where global rotations are expensive and may amplify outliers. Importantly, since regular $\mathcal{H}$-Matrices minimize column discrepancy, they effectively mitigate row-wise outliers introduced by Sylvester-type constructions.

## 4 METHOD

In this work, we present an plug-and-play W4A4 quantization method for Diffusion Transformers (DiTs). As illustrated in Figure 4, Our approach consists of two core components: **ConvRot** and **ConvLinear4bit**.ConvRot performs group-wise regular Hadamard rotations, where the group-wise design reduces computational cost and alleviates row-wise aggregation, while the regular structure further resolves the row-wise issue, smoothing activation and weight distributions before quantization. ConvLinear4bit fuses ConvRot, 4-bit quantization, matrix multiplication, and dequantization into a single linear layer, enabling straightforward plug-and-play replacement of the original layers. By simply replacing the original linear layers with ConvLinear4bit, we can perform low-precision inference on large-scale DiT models while maintaining high visual quality.

### 4.1 MOTIVATION

Our first motivation comes from the high cost of existing rotation-based quantization methods, such as QuaRot (Ashkboos et al., 2024) and SpinQuant (Liu et al., 2024), which apply a global Hadamard rotation. This redistributes outliers across all channels but incurs quadratic complexity, making it expensive for large diffusion models. We reduce this cost by limiting rotations to smaller groups of size $N_0$, balancing computation and outlier smoothing.

The second motivation arises from row-wise outliers in activations (see Figure 3). Large Hadamard matrices with high-magnitude columns can concentrate such outliers, increasing their values instead

of smoothing them. Group-wise rotations limit this effect, and regular Hadamard matrices ensures minimal column discrepancy, smoothing activations while avoiding row-wise outlier aggregation.

These two insights motivate the design of ConvRot: (i) reducing computational cost by restricting the rotation scope, and (ii) preventing row-wise outlier concentration through regular Hadamard matrices.

### 4.2 CONVROT: GROUP-WISE REGULAR HADAMARD ROTATION

As discussed in Section 3.1, inserting a Hadamard transformation within a matrix multiplication can be interpreted as rotating the input or weight space without changing the output distribution. Building upon this, we propose ConvRot, which applies a **group-wise Regular Hadamard Transform (RHT)** to control the scope of rotation and improve both computational efficiency and outlier handling.

Existing rotation-based quantization methods, such as QuaRot and SpinQuant, typically apply a global Hadamard transform of order $K$, which incurs quadratic complexity $\mathcal{O}(K^2)$ since outliers are redistributed across all channels simultaneously. This becomes prohibitive for large-scale diffusion models. Our key insight is that by partitioning the feature dimension into blocks of size $N_0$ and applying a **Regular Hadamard Transform** within each block, we can reduce the computational cost and localize outlier redistribution.

Formally, given a standard linear layer

$$\mathbf{Y} = \mathbf{X}\mathbf{W}^\top, \quad \mathbf{X} \in \mathbb{R}^{M \times K}, \mathbf{W} \in \mathbb{R}^{N \times K}, \tag{11}$$

we partition the input and weight matrices into column-wise blocks of size $N_0$:

$$\mathbf{X} = [\mathbf{X}_1, \mathbf{X}_2, \dots, \mathbf{X}_{\lceil K/N_0 \rceil}], \quad \mathbf{W} = [\mathbf{W}_1, \mathbf{W}_2, \dots, \mathbf{W}_{\lceil K/N_0 \rceil}]. \tag{12}$$

For each block, we insert a Regular Hadamard Rotation (RHT):

$$\mathbf{Y} = \sum_{i=1}^{\lceil K/N_0 \rceil} \mathrm{RHT}(\mathbf{X}_i)\, \mathrm{RHT}(\mathbf{W}_i)^\top. \tag{13}$$

By performing group-wise RHT, we reduce computational complexity from $\mathcal{O}(K^2)$ to $\mathcal{O}(K)$, while preserving effective outlier suppression. Importantly, the equivalence property ensures that this local rotation does not change the overall linear transformation; it only redistributes the information within each block, providing finer-grained control over activation distributions. Figure 5 shows peak activation values and accuracy-speed trade-offs under different group sizes $N_0$.

It is important to note that existing FWHT-based implementations achieve speedup via FFT-like butterfly operations, but this restricts them to CUDA cores and prevents leveraging Tensor Cores. In contrast, we implement ConvRot using matrix multiplication, which avoids extra memory movement and fully exploits highly optimized matmul pipelines on modern GPUs, yielding significant speedups over FWHT. Formally, a group rotation of size $N_0$ can be implemented as a convolution-like operation on the input activation with kernel size $[1, N_0]$, channels = out channels = $N_0$, and stride = $(1, N_0)$, motivating the name **ConvRot** as it fuses the concepts of convolution and rotation.

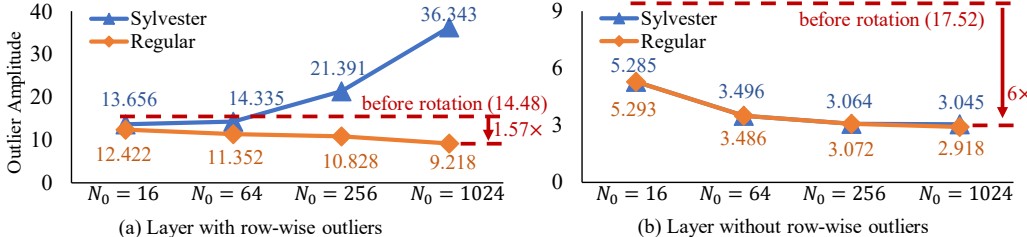

(a) Layer with row-wise outliers      (b) Layer without row-wise outliers

Figure 5: Impact of rotation matrix type and size on activation outliers. We compare Regular and Sylvester Hadamard matrices under two scenarios: with and without row-wise outliers. Results show that, in the presence of row-wise outliers, increasing rotation size $N_0$ with Regular Hadamard matrices effectively suppresses them, whereas Sylvester matrices tend to amplify them. Without row-wise outliers, both types perform similarly.

### 4.3 CONVLINEAR4BIT

Building on ConvRot, we develop **ConvLinear4bit**, which allows straightforward replacement of original linear layers for plug-and-play 4-bit inference as shown in Figure 4. ConvLinear4bit fuses ConvRot, quantization, 4-bit matrix multiplication, and dequantization into a single layer. For ConvRot, we implement the group-wise Hadamard rotation via reshape-based matrix multiplication, completely avoiding memory movement. The quantization, 4-bit matrix multiplication, and dequantization operations follow the design in QuaRot (Ashkboos et al., 2024) and utilize highly optimized CUDA kernels to take advantage of GPUs' int4 Tensor Cores.

In summary, ConvRot provides a flexible mechanism to control the number of channels participating in outlier redistribution, trading off computation and smoothing effectiveness. Matmul-based implementation allows leveraging modern GPU pipelines efficiently, while ConvLinear4bit enables practical, plug-and-play 4-bit weight-activation quantization for large-scale diffusion models, achieving significant memory reduction and inference speedup without sacrificing image quality.

## 5 EXPERIMENTS

### 5.1 SETUPS

**Models.** We conduct our experiments on **FLUX.1.DEV** (Black-Forest-Labs, 2024), which is a 12B-parameter text-to-image diffusion model known for its high-quality image generation capabilities. Typically, inference with this model requires over 30GiB of GPU memory, making it hard to deploy on consumer-grade hardware.

**Datasets.** Following Li et al. (2024b), we evaluate generation quality on a subset of 5K prompts stratified sampled across categories from the **MJHQ-30K** dataset (Li et al., 2024a), a benchmark curated from Midjourney containing 30K high-quality images in 10 diverse categories, filtered by aesthetic and CLIP scores to ensure strong visual quality and text alignment.

**Baselines.** We compare ConvRot against the following baselines:

- SVDQuant (Li et al., 2024b) is a state-of-the-art 4-bit quantization framework for diffusion models that combines a novel low-rank branch design with a carefully co-optimized inference engine, achieving much higher speedups than standard quantization methods through engineering optimizations beyond quantization itself.

- QuaRot (Ashkboos et al., 2024) is a representative rotation-based quantization method for LLMs that reduces the number of rotations by fusing them into adjacent weight matrices and employs fast Walsh-Hadamard transforms (FWHT) to minimize rotation latency. However, 1) the fusion technique cannot be applied to FLUX due to the presence of adaptive normalization layers (Peebles & Xie, 2023), and 2) the FWHT rotation matrix is equivalent to the standard Hadamard matrix, which can lead to row-wise outlier clustering. Therefore, we only use it to compare single-layer outlier suppression and acceleration effects.

**Metrics.** We conduct a comprehensive evaluation at both the single-layer and end-to-end levels. For single linear layer analysis, we evaluate **Precision** by measuring the post-rotation outlier amplitude and the maximum layer error. We also assess **Efficiency** by benchmarking the rotation latency and overall layer latency. For the end-to-end text-to-image task, we evaluate from two aspects: **Quality**, using FID (Heusel et al., 2017) (lower is better) and ImageReward (IR) (Xu et al., 2024) (higher is better); and **Similarity** to the original BF16 model, using LPIPS (lower is better) and PSNR (higher is better). We also report DiT memory footprint and end-to-end generation latency.

**Implementation details.** For ConvRot, we use per-token/per-channel 4-bit quantization with regular Hadamard rotations, denoting ConvRot-$N_0$ as group size $N_0$. Regular denotes regular Hadamard matrices, standard denotes Sylvester-type Hadamard, and random denotes random orthogonal matrices. Layers with larger outlier amplitudes use larger rotations ($N_0 = 1024$) to better smooth outliers. To preserve fine-grained details, `proj_out` remains in FP16 (following QuaRot's implementation). SVDQuant and QuaRot use their official defaults. Experiments run on a single RTX 4090 (24GB), with CPU offloading for models exceeding GPU memory.

Table 1: Precision and efficiency of different rotation implementations in a single Flux linear layer (in=15360, out=3072) with the prompt "A cute cat." QuaRot uses a Sylvester-type Hadamard of size 15360, realized by matrix multiplication + FWHT. Sylvester/FWHT and Regular/Group-wise RHT represent two H-Matrix Types.

| Rotation Setting | | Precision | | Efficiency (ms) | |
|---|---|---|---|---|---|
| Type | Size | Outlier Amp. ↓ | Max Err. ↓ | Rot. Lat. ↓ | Layer Lat. ↓ |
| FP16 | — | 18.00 | — | — | 1.943 |
| QuaRot | 15360 | 105.63 | 1306.00 | 0.690 | 1.795 |
| Sylvester/FWHT | 16 | 13.66 | 27.25 | 3.782 | 4.586 |
| | 64 | 14.34 | 27.50 | 1.066 | 2.136 |
| | 256 | 21.39 | 25.56 | 0.446 | 1.588 |
| | 1024 | 36.34 | 338.00 | 0.465 | 1.623 |
| Regular/GrpRHT | 16 | 12.42 | 36.13 | 0.364 | 1.424 |
| | 64 | 11.35 | 25.91 | **0.252** | **1.352** |
| | 256 | 10.83 | 32.13 | 0.277 | 1.394 |
| | 1024 | **9.22** | **23.03** | 0.729 | 1.843 |

Table 2: End-to-end performance comparison on FLUX.1-dev (50 steps). Lower LPIPS/FID and higher PSNR/IR/Human indicate better performance. SVDQuant's method maintains a parallel 16-bit LoRA branch, while our rollback strategy selectively keeps a few highly sensitive layers in 16-bit precision.

| Method | Precision | DiT Memory | Latency | Quality | | Similarity | |
|---|---|---|---|---|---|---|---|
| | | (GiB) | (s) | FID ↓ | IR ↑ | LPIPS ↓ | PSNR ↑ |
| Baseline | BF16 | 22.7 | 54.6 | 10.07 | 0.992 | - | - |
| SVDQuant | W4A4 + BF16$_{LoRA}$ | 6.5 | 14.9 | 10.01 | 0.974 | 0.182 | 21.24 |
| ConvRot-16 | W4A4 | 5.6 | 23.2 | 14.27 | 0.596 | 0.247 | 18.60 |
| + FP16$_{proj\_out}$ | W4A4 + FP16 proj_out | 5.8 | 24.3 | 11.81 | 0.637 | 0.246 | 19.24 |
| ConvRot-64 | W4A4 | 5.6 | 22.6 | 13.25 | 0.735 | 0.242 | 18.28 |
| + FP16$_{proj\_out}$ | W4A4 + FP16 proj_out | 5.8 | 23.9 | 11.25 | 0.833 | 0.220 | 20.05 |
| ConvRot-256 | W4A4 | 5.6 | 23.0 | 12.19 | 0.822 | 0.211 | 20.36 |
| + FP16$_{proj\_out}$ | W4A4 + FP16 proj_out | 5.8 | 24.2 | 10.33 | 0.961 | 0.185 | 21.17 |

## 5.2 SINGLE-LAYER ANALYSIS

Table 1 compares precision and efficiency of different rotation implementations in a single Flux linear layer (in=15360, out=3072) with the prompt "A cute cat.". Outlier Amplitude denotes the maximum absolute activation, Max Error the largest element-wise deviation from the FP16 layer, Rotation Latency the rotation runtime, and Layer Latency the full layer runtime. As shown in Figure 3, layers such as `single_transformer_blocks.proj_out` and

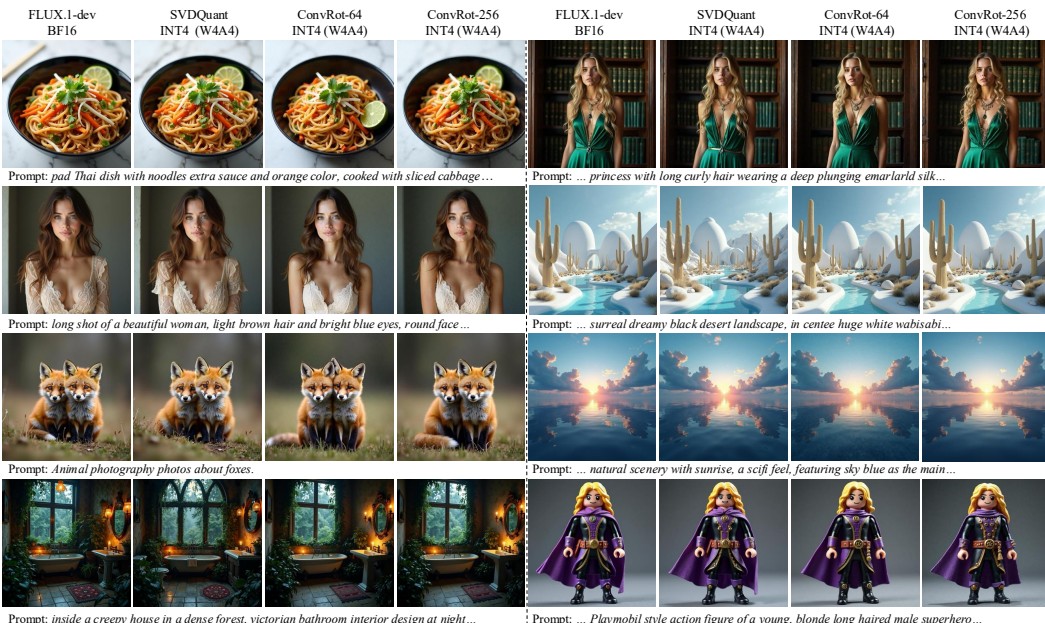

Figure 6: Visual comparison on the MJHQ-30K dataset. The prompts cover diverse themes including food, human portraits, animals, landscapes, indoor scenes, and figurines.

`transformer_blocks.ff_context.net.2` exhibit pronounced row-wise outliers, with full-size Sylvester-type Hadamard reaching Outlier Amplitude 105.63 and Max Error 1306.00.

For the FWHT implementation, small group sizes (e.g., $N_0 = 16$) require many rotation calls, increasing latency, while larger $N_0$ aggregates row-wise outliers due to the first column of all ones, reducing precision. In contrast, Group-wise RHT shows decreasing Outlier Amplitude and Max Error with larger $N_0$, effectively mitigating row-wise outliers. Rotation latency also decreases with $N_0$, reaching $1.43\times$ speedup over FP16 at $N_0 = 64$.

### 5.3 END-TO-END PERFORMANCE ON TEXT-TO-IMAGE GENERATION

We compare our ConvRot approach with SVDQuant and the BF16 baseline on FLUX.1-dev with 50 sampling steps. As summarized in Table 2, we evaluate both **Similarity** (LPIPS, PSNR) and **Quality** (FID, IR). Lower LPIPS/FID scores and higher PSNR/IR scores indicate better performance. ConvRot significantly reduces both DiT memory and latency compared to the BF16 baseline. Metrics indicate that INT4 quantization causes more degradation in image quality than in structural similarity. This is because the quantized model has a limited numerical representation space, making it difficult to preserve high-frequency details in the generated images, discussed in Section 5.4.

To mitigate this degradation, we selectively maintain a few highly sensitive layers in FP16 precision. This rollback strategy effectively restores fine-grained details in the generated images, improving perceptual quality while retaining most of the efficiency gains of INT4 inference. As shown in Table 2, ConvRot with partial FP16 rollback achieves a favorable balance between memory, latency, and image fidelity, demonstrating its practical effectiveness for large-scale text-to-image generation.

### 5.4 ABLATION STUDY

We further investigate the impact of group size $N_0$ and rotation type on image generation quality. Unlike the experiments in Section 5.3, here we apply the same $N_0$ uniformly across all layers. This setup isolates the effect of rotation configuration on the model's representational capacity and image fidelity. In our results, the regular rotation with $N_0 = 256$ consistently achieves the best performance. The standard rotation performs well at $N_0 = 16$ or $64$, but exhibits degradation at $N_0 = 256$. The random rotation shows unstable performance across different random seeds, yet it does not suffer from the degradation observed when $N_0$ increases. Results are provided in the Appendix B.

We observe that, without special treatment, the outputs of a quantized model often exhibit banding or granularity artifacts. This is because the INT4 quantization severely limits the model's expressive capacity, preventing it from producing smooth color transitions. Under effective outlier suppression, the quantized model tends to degrade more noticeably in low-frequency details, such as gradual color gradients, while still preserving overall structural integrity and high-frequency details, such as textures in hair, noodles, or flower patterns. A promising direction to mitigate these limitations is to enhance the model's expressive power via high-precision LoRA branches, or by keeping a subset of layers or sampling steps in higher precision.

## 6 CONCLUSION

We present ConvRot, a novel rotation-based quantization framework for diffusion transformers that enables efficient W4A4 INT4 inference while preserving image quality. To address the challenges of row-wise and column-wise outliers in activations, we introduce a group-wise rotation scheme based on regular Hadamard matrices, reducing computational complexity from quadratic to linear complexity and significantly lowering rotation latency compared to global rotations. Building on this, we design ConvLinear4bit, a plug-and-play module that fuses rotation, quantization, GEMM, and dequantization, allowing all linear layers in a diffusion model to be quantized without retraining. Extensive experiments on FLUX.1-dev demonstrate that our approach stably suppresses outliers (up to $7\times$), reduces memory usage by $4.05\times$, and achieves a $2.26\times$ speedup, while maintaining high-fidelity image generation. To our knowledge, this work is the first to apply rotation-based quantization to diffusion transformers for fully INT4 W4A4 inference, providing a practical solution for accelerating large-scale text-to-image generation with minimal quality loss.

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

## A  PROOFS

### A.1  PROOF OF THEOREM 3.1 (COLUMN SUM SQUARED PROPERTY)

*Proof.* Let $\mathbf{H}_n$ be a Hadamard matrix of order $n$, satisfying $\mathbf{H}_n \mathbf{H}_n^\top = n\mathbf{I}_n$. Define the column sums as $c_j = \sum_{i=1}^{n} H_{ij}$ for $j = 1, \ldots, n$. We consider the squared $\ell_2$ norm of the vector of column sums:

$$\sum_{j=1}^{n} c_j^2 = \sum_{j=1}^{n} \left( \sum_{i=1}^{n} H_{ij} \right)^2. \tag{14}$$

This can be expressed as

$$\sum_{j=1}^{n} c_j^2 = \|\mathbf{H}_n^\top \mathbf{1}\|_2^2 = \mathbf{1}^\top \mathbf{H}_n \mathbf{H}_n^\top \mathbf{1}. \tag{15}$$

Using the orthogonality property $\mathbf{H}_n \mathbf{H}_n^\top = n\mathbf{I}_n$, we obtain

$$\mathbf{1}^\top \mathbf{H}_n \mathbf{H}_n^\top \mathbf{1} = n \cdot \mathbf{1}^\top \mathbf{1} = n \cdot n = n^2. \tag{16}$$

Therefore,

$$\sum_{j=1}^{n} \left( \sum_{i=1}^{n} H_{ij} \right)^2 = n^2, \tag{17}$$

which proves the claim. $\qquad\square$

### A.2 PROOF OF THEOREM 3.2 (COLUMN DISCREPANCY OF REGULAR HADAMARD)

**Lemma A.1** (Column Sum of Regular Hadamard Matrix). *If $\mathbf{H}_n$ is a regular Hadamard matrix, then each column sum satisfies*

$$\sum_{i=1}^{n} H_{ij} = \pm\sqrt{n}, \quad \forall j = 1, \ldots, n. \tag{18}$$

*Proof.* By definition, a regular Hadamard matrix has all row and column sums equal in magnitude to $\sqrt{n}$. Let $\mathbf{H}_n$ be regular. Then by the row-column symmetry of Hadamard matrices (orthogonality and $\pm 1$ entries), each column must sum to the same absolute value as the row sum. Since the squared sum of all columns is

$$\sum_{j=1}^{n} \left( \sum_{i=1}^{n} H_{ij} \right)^2 = n^2,$$

and there are $n$ columns, each column sum squared must equal $n$, i.e.,

$$\left( \sum_{i=1}^{n} H_{ij} \right)^2 = n \implies \sum_{i=1}^{n} H_{ij} = \pm\sqrt{n}.$$

$\qquad\square$

*Proof of Theorem 3.2.* By definition, the column discrepancy of $\mathbf{H}_n$ is

$$\|\mathbf{H}_n^\top \mathbf{1}\|_\infty = \max_{1 \le j \le n} \left| \sum_{i=1}^{n} H_{ij} \right|. \tag{19}$$

Applying Lemma A.1, each column sum of a regular Hadamard matrix is $\pm\sqrt{n}$. Hence, the maximum absolute column sum is exactly $\sqrt{n}$.

Since $\sqrt{n}$ is also the theoretical minimum discrepancy achievable by any Hadamard matrix, regular Hadamard matrices attain the optimum. $\qquad\square$

### A.3 PROOF OF THEOREM 3.3 (KRONECKER CONSTRUCTION OF REGULAR HADAMARD MATRICES)

*Proof.* We prove by induction on $k$.

**Base case:** For $k = 1$, the given $4 \times 4$ matrix $\mathbf{H}_4$ is regular, since each row and column sums to $\pm 2 = \pm\sqrt{4}$.

**Inductive step:** Assume $\mathbf{H}_{4^k}$ is a regular Hadamard matrix, i.e., each row and column sums to $\pm\sqrt{4^k}$. Consider $\mathbf{H}_{4^{k+1}} = \mathbf{H}_{4^k} \otimes \mathbf{H}_4$. For any column of $\mathbf{H}_{4^{k+1}}$, the Kronecker product structure

ensures that its entries are composed of four blocks, each proportional to a column of $\mathbf{H}_{4^k}$. Therefore, the column sum is

$$\sum_i H_{ij}^{(4^{k+1})} = \left(\sum_u H_{u,v}^{(4^k)}\right) \cdot \left(\sum_w H_{w,z}^{(4)}\right), \tag{20}$$

where $v, z$ index the corresponding columns in $\mathbf{H}_{4^k}$ and $\mathbf{H}_4$. By the induction hypothesis, $\sum_u H_{u,v}^{(4^k)} = \pm\sqrt{4^k}$, and since $\mathbf{H}_4$ is regular, $\sum_w H_{w,z}^{(4)} = \pm 2$. Thus,

$$\sum_i H_{ij}^{(4^{k+1})} = (\pm\sqrt{4^k})(\pm 2) = \pm\sqrt{4^{k+1}}. \tag{21}$$

Hence $\mathbf{H}_{4^{k+1}}$ is also regular. By induction, a regular Hadamard matrix exists for all $n = 4^k$. $\square$

## B  MORE RESULTS

Here, the ConvRot-$N_0$ results correspond to the model without any FP16 layers preserved and without applying special large-scale rotations to sensitive layers, allowing a clearer observation of how the rotation matrix affects the degradation of generated image quality.

FLUX.1-dev BF16     ConvRot 16 W4A4     ConvRot 64 W4A4     ConvRot 256 W4A4     ConvRot 256 + Rollback

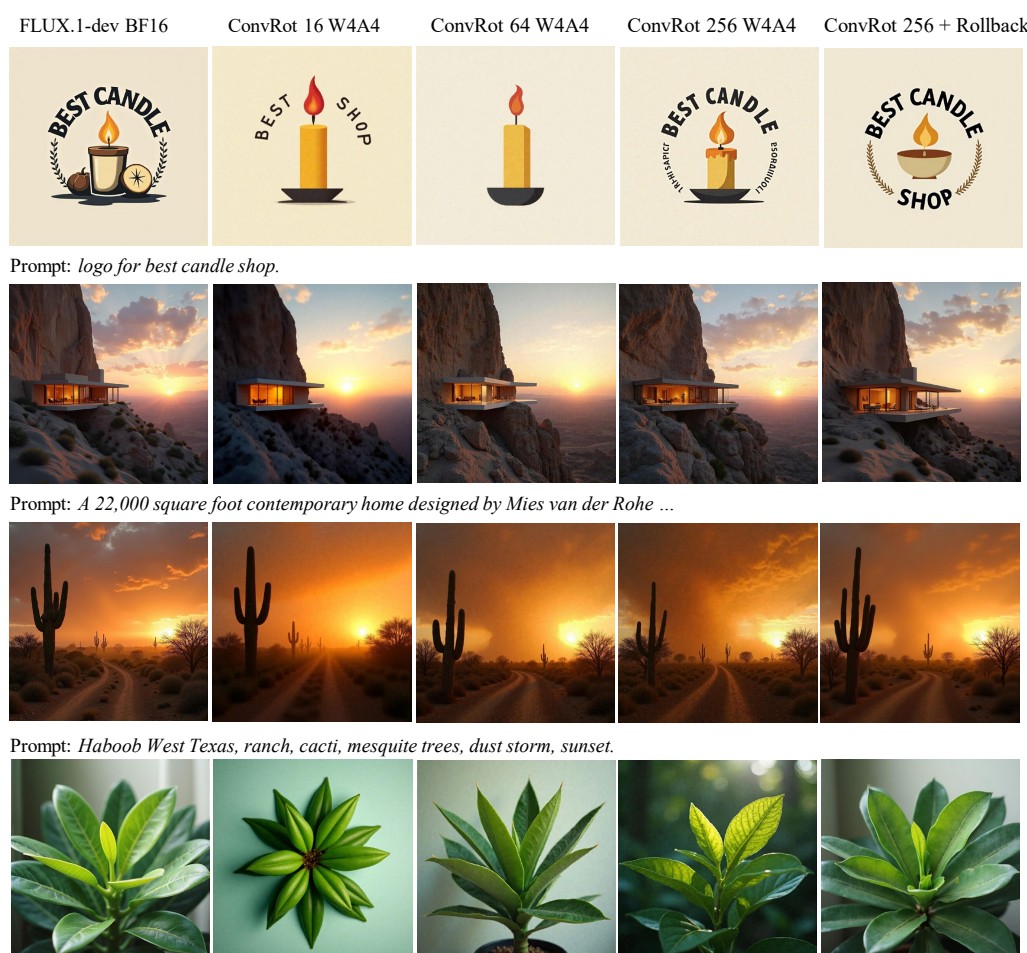

Prompt: *logo for best candle shop.*

Prompt: *A 22,000 square foot contemporary home designed by Mies van der Rohe ...*

Prompt: *Haboob West Texas, ranch, cacti, mesquite trees, dust storm, sunset.*

Prompt: *photo of a ficus, shot on Afga Vista 400, flat lighting.*

Figure 7: Ablation study on $N_0$ (Part 1).

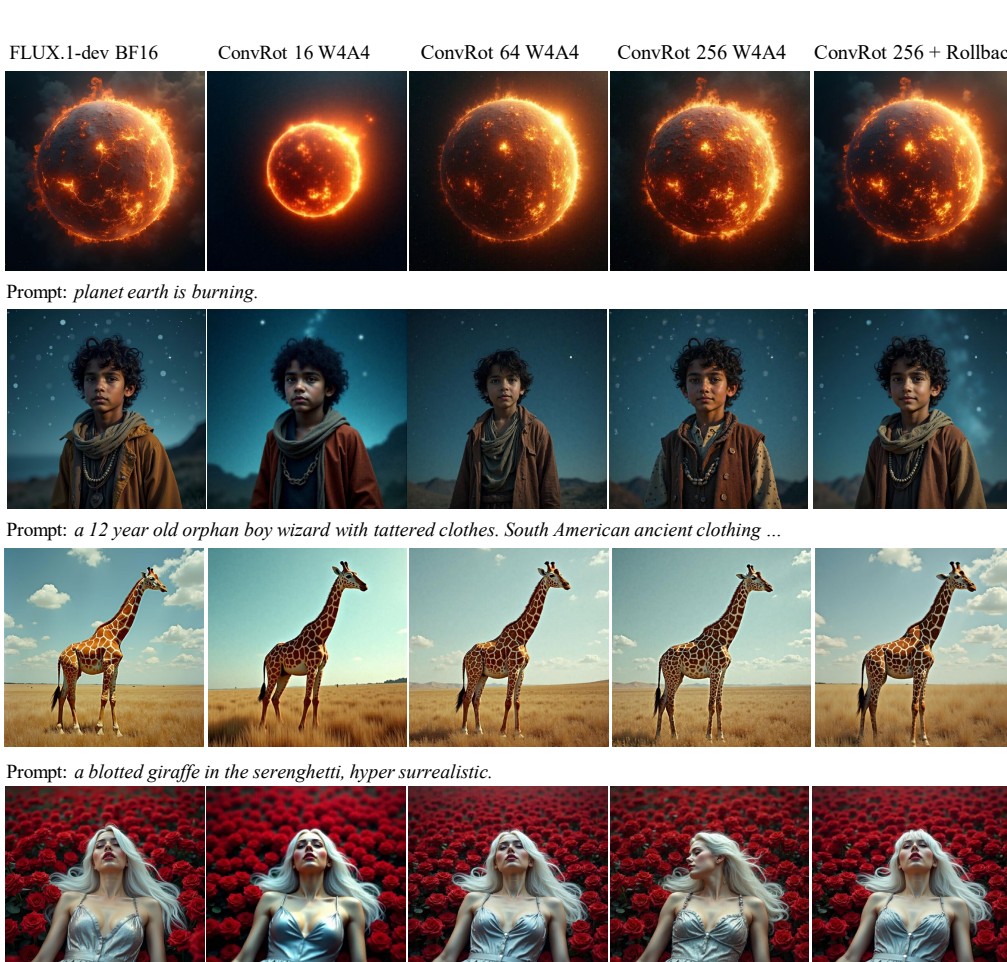

FLUX.1-dev BF16    ConvRot 16 W4A4    ConvRot 64 W4A4    ConvRot 256 W4A4    ConvRot 256 + Rollback

Prompt: *planet earth is burning.*

Prompt: *a 12 year old orphan boy wizard with tattered clothes. South American ancient clothing ...*

Prompt: *a blotted giraffe in the serenghetti, hyper surrealistic.*

Prompt: *Extreme LowAngle. a woman with long eggshell white hair lying among a sea of wet red roses in a silver ...*

Figure 8: Ablation study on $N_0$ (Part 2).

FLUX.1-dev BF16    ConvRot 16 W4A4    ConvRot 64 W4A4    ConvRot 256 W4A4    ConvRot 256 + Rollback

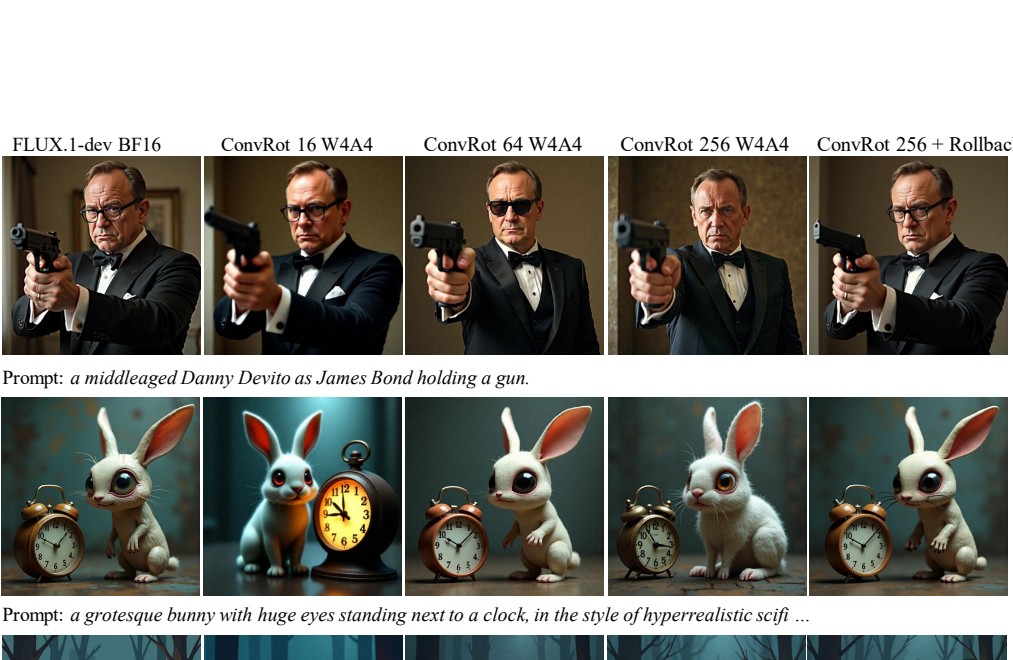

Prompt: *a middleaged Danny Devito as James Bond holding a gun.*

Prompt: *a grotesque bunny with huge eyes standing next to a clock, in the style of hyperrealistic scifi ...*

Prompt: *dnd illustration of a small cute panda playing a lyre in the woods next to a cozy campfire.*

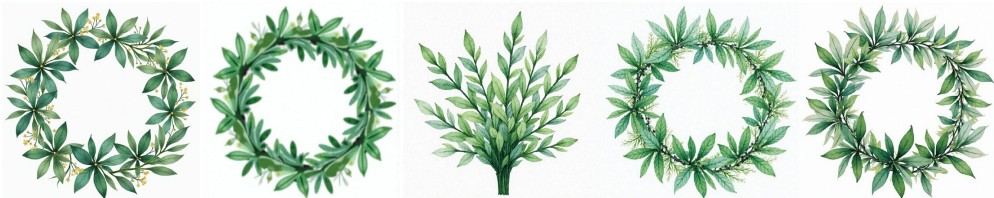

Prompt: *Greenery Watercolor Clipart, Greenery Bundle Clipart,Botanical Clipart,Watercolor Clipart,Wedding Clipart, ...*

Figure 9: Ablation study on $N_0$ (Part 3).

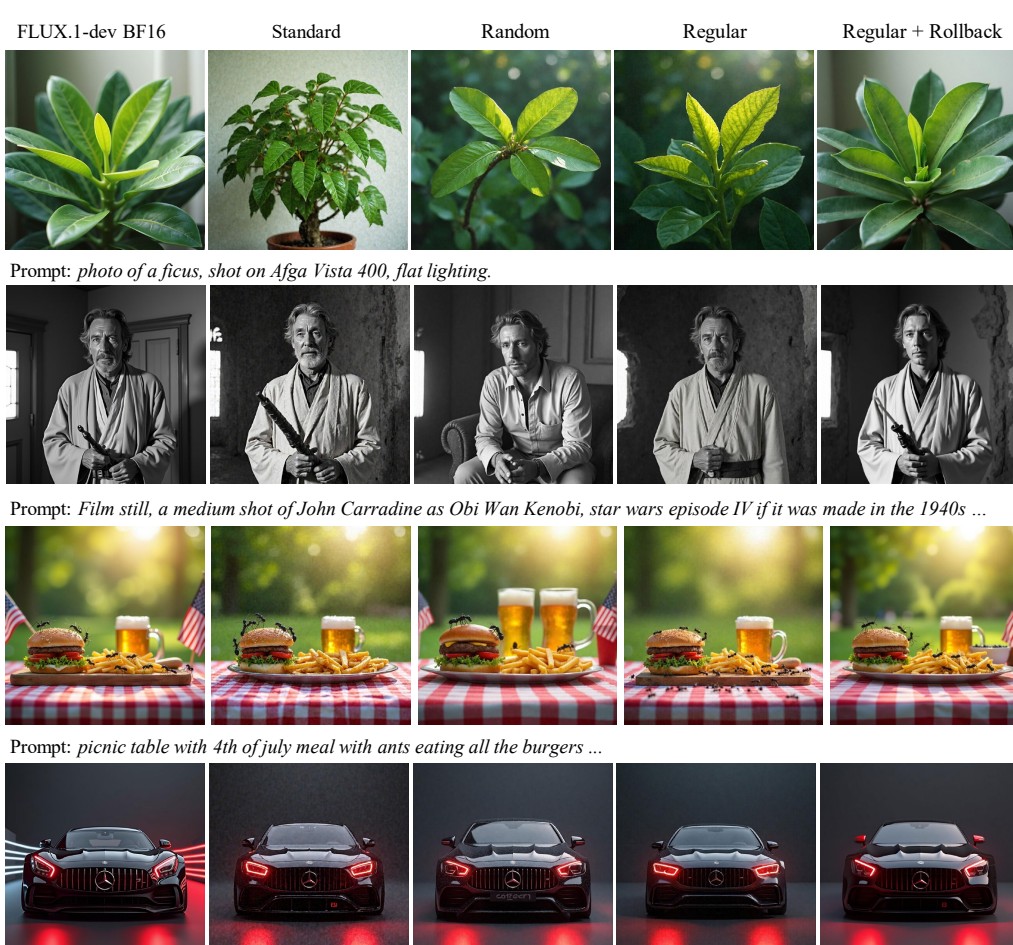

| FLUX.1-dev BF16 | Standard | Random | Regular | Regular + Rollback |

Prompt: *photo of a ficus, shot on Afga Vista 400, flat lighting.*

Prompt: *Film still, a medium shot of John Carradine as Obi Wan Kenobi, star wars episode IV if it was made in the 1940s ...*

Prompt: *picnic table with 4th of july meal with ants eating all the burgers ...*

Prompt: *ultra photo realistic, ultra detailed, neon, black and red 2023 AMG mercedes benz, octane render, 8k.*

Figure 10: Ablation study on Rotation Types.

