# OpenReview forum: "ConvRot: Rotation-Based Plug-and-Play 4-bit Quantization for Diffusion Transformers"
_ICLR.cc/2026/Conference — ICLR 2026 Conference Withdrawn Submission_

### Official Review · Reviewer_XHML · 2025-10-14

**Soundness:** 2
**Presentation:** 4
**Contribution:** 2
**Rating:** 2
**Confidence:** 5

**Summary:**

The paper proposes ConvRot. ConvRot applies blocked Hadamard-like transforms to improve activation quantization in DiT models. In contrast to prior works, they construct the Hadamard transform so that the sum per column/row is constant, $\pm\sqrt(d)$. This is valuable, as it avoids having a column (as done the standard Hadamard Transform) with only 1s---the latter creates an outlier if there is a non-zero mean in the channels.

**Strengths:**

I like the paper overall! Some strengths:
* The presentation of the paper is strong, with clear motivation, illustrations, and a clear method.
* The idea of deviating from the standard FHT by including the $H_4$ (Eq. 10) term is novel and useful. 0th channel outliers due to a Naive Hadamard transform is a real problem and addressing this directly makes sense to me.
* The results are promising for a method without calibration/QAT.

**Weaknesses:**

**1. Baselines.**

My major concern are the limited results.
(a) Arguably the closest baselines, QuaRot or FWHT applied to a smaller size (e.g. 256), is not compared against in the end-to-end results of Table 2. Recent transformation literature, e.g. FlatQuant, OstQuant, HadaNorm, are also all missing, and may well do better than ConvRot.
(b) I'm also a bit doubtful about the speed comparison w.r.t. FWHT---why would RHT be any faster?

**2. Rollout.**

The authors mention they use "rollout", i.e. keeping some sensitive layers in high-precision. Without this rollout, ConvRot seems to perform poorly. It is unclear how e.g. a method like QuaRot/FlatQuant would also benefit from this, consequently it is not clear whether ConvRot would outperform these methods if they were compared fairly. In particular, the poor QuaRot results of Table 1 are represented for exactly the layers that rollout keeps in high-precision anyway.

**3. Computational cost**

> Our first motivation comes from the high cost of existing rotation-based quantization methods, such
as QuaRot (Ashkboos et al., 2024) and SpinQuant (Liu et al., 2024), which apply a global Hadamard
rotation. This redistributes outliers across all channels but incurs quadratic complexity

FHT may be quadratic in the number of Hadamard levels $K$, but since $K=\log_2(dim)$, $K$ hardly grows for larger models. All in all, I think the theoretical cost of FHT is already marginal compared to matmuls, and any additional *theoretical* gains are even more marginal. In any case, the idea of using smaller blocks has been explored in prior work (e.g. DuaRot at NeurIPS 2024)

___

Overall, I see true value in the paper, but the current results do not provide enough evidence for it. I'm willing to improve my score if 1 and 2 are addressed in detail.

**Questions:**

The problem of the Sylvester-type HT is a non-zero mean in the channels, leading to the $\mathbf{1}$ column approximating `mean*sqrt(d)`. In Figure 3, the top-right figure seems to have a one-sided distribution for 3/4 of the channels, hence the mean would indeed be non-zero. In general, I assume many distributions do not have such off-distribution and may be quite zero-centered. Do you agree? If so, does this have implications on where ConvRot may improve over FWHT?

In any case, we could zero-center it by multiplying the channels by -1,1 (e.g. as done in FlatQuant and OstQuant), which could also approximately zero-center the channels.

---

### Official Review · Reviewer_7z1x · 2025-10-22

**Soundness:** 3
**Presentation:** 2
**Contribution:** 2
**Rating:** 4
**Confidence:** 4

**Summary:**

The paper introduces ConvRot: a simple yet effective strategy to enable 4 bits activation and weight quantization on FLUX1-dev. ConvRot introduces 2 major changes to the standard Hadamard transformations used for activation and weight quantization: 1) instead of using Sylvester Hadamard constructions, the paper motivates the use of Regular Hadamard matrices, and 2) Instead of using the FWHT algorithm, Hadamard rotations are applied using a convolutional operation. This strategy results in a substantial memory usage reduction and speed-up in the FLUX1-dev architecture without requiring any calibration or re-training procedure.

**Strengths:**

* The paper proposes a simple yet elegant method to deal with issues introduced by Hadamard rotations for activation and weight quantization.

* The paper introduces a simple Plug and Play method that can effectively reduce the memory and compute requirements of LVMs without substantially changing the architectures nor requiring a calibration/retraining procedure.

**Weaknesses:**

* The paper uses Outlier Amplitude as a metric to measure the effectiveness of the proposed method. However, the metric is not clearly introduced or motivated.

* The comparison with QuaRot is not exhaustive since the benefits of using (block) Regular Hadamard instead of (block) Sylvester Hadamard are not compared directly in terms of end-to-end performance.

* Some of the computational considerations behind the choice of the use of a convolutional matrix multiplication instead of the FHWT algorithms are not clear nor well motivated. In particular, Sylvester (block) Hadamard matrices can also be applied using convolutional operations. Hence, the paper could do a better job at isolating latency improvements by comparing FWHT and Convolutional matrix multiplication directly. Furthermore, the empirical increase in latency reported for FWHT on small matrices is not clearly explained.

* The paper reports experiments on one LVM only. Even if similar trends are to be expected on different LVMs, additional results can strengthen the submission.

**Questions:**

1. Line 282-284 the paper mentions that global Hadamard transform of order $K$ incurs in quadratic complexity. However, the paper also mentions that FWHT has complexity $K\log K$. Can the authors clarify the reason behind this inconsistency? Is this related to the use of Tensor Cores? Can the authors clarify why butterfly-type operations are not suitable for Tensor Cores?

2. Can the authors further clarify the statement on lines 301-302? Why does the computational complexity decrease from $K^2$ to $K$ by performing group-wise operations? Shouldn’t this hold also for the Sylvester construction?

3. How does the proposed outlier magnitude metric relate to the quantization error? Do the values of outlier magnitude reported in Table 1 refer to the activations or the weights? Are the values averaged across multiple rows/columns or do they refer to the maximum over the whole matrix/tensor?

4. Why does the latency for FWHT increase with decreasing block size (Table 1)? Shouldn’t all block rotations be performed in parallel?

5. The same convolutional style-operation proposed in this paper can also be applied to the Sylvester Hadamard construction. In this scenario, the latency would be identical, so the benefit of ConvRot should come from the use of Regular instead of Sylvester Hadamard matrices. The benefit of the former option is not clear since outliers and maximum error seem lower only at large block size, at which point FWHT is faster. Furthermore, there is no end2end performance comparison/ablation between the two versions in Table 2. How does (block) QuaRot compare against ConvRot for the setups reported in Table 2?

---

### Official Review · Reviewer_hUxq · 2025-10-29

**Soundness:** 2
**Presentation:** 2
**Contribution:** 2
**Rating:** 2
**Confidence:** 3

**Summary:**

This paper introduces ConvRot, a training-free W4A4 quantization method for diffusion transformers. The method uses a principled, Kronecker-based Hadamard rotation to balance weights, guided by a novel discrepancy metric. For efficiency, ConvRot employs group-wise rotations and a fused `ConvLinear4b` kernel, achieving up to a 2.26× speedup and 4.05× memory reduction on models like FLUX-1-dev with minimal impact on image quality.

**Strengths:**

1. Practical design with group-wise rotations that reduce computational cost and enable a plug-and-play, training-free W4A4 quantization scheme.
2. The method does not require a calibration dataset, which simplifies implementation.
3. Systems-level contribution via the `ConvLinear4b` kernel, bridging algorithmic design with practical engineering to foster adoption.

**Weaknesses:**

1. The evaluation scope is limited, focusing primarily on FLUX-1-dev. Broader testing across more diffusion transformer variants would be necessary to establish the generality of the method.
2. The paper lacks a detailed, operator-level breakdown of performance. A thorough analysis of time and memory overheads for rotation, quantization/dequantization, and GEMM would provide a clearer picture of the practical costs.
3. The paper does not sufficiently contrast the proposed technique with existing methods, such as SVDQuant, making it difficult to assess its relative advantages and contributions.

**Questions:**

1. What criteria are used to determine which layers are kept in FP16/BF16 precision and which are quantized?
2. Could you provide results on additional models to better demonstrate the method's generality?
3. How does the method perform with FP4 quantization, and would any modifications be required?

---

### Official Review · Reviewer_TuoY · 2025-11-03

**Soundness:** 2
**Presentation:** 2
**Contribution:** 2
**Rating:** 4
**Confidence:** 3

**Summary:**

This paper identifies a drawback of previous rotation-based quantization methods that could amplify row-wise outliers in certain circumstances. They propose a group-wise rotation-based quantization algorithm that uses a regular Hadamard Transform to smooth both row-wise and column-wise outliers and reduce the computational complexity. They also implement a kernel to fuse rotation, quantization, GEMM, and dequantization, and demonstrate a 2.26x speedup over the BF16 baseline.

**Strengths:**

- This work addresses a known failure mode of global Hadamards in DiTs.
- The quantization method is demonstrated on real GPUs and can provide end-to-end speedup.

**Weaknesses:**

- The quality and performance metrics still fall behind the state-of-the-art quantization method, SVDQuant.
- Although they claimed their method could smooth both row-wise and column-wise outliers, the evaluation results still show a significant quality drop if proj_out is quantized to 4 bits, limiting the usefulness of the proposed method.
- Only one workload (FLUX.1-dev) is used to evaluate the results.
- Code is not provided.

**Questions:**

- Please provide an analysis of the performance and quality when comparing to SVDQuant. Is this an engineering issue or a fundamental limitation of the proposed algorithm? Can QuaRot combine with SVDQuant to provide higher performance or quality?
- How does the proposed method work on newer quantization schemes (like NVFP4 on latest generation GPUs)?
- How does the kernel fusion of quantization and 4-bit GEMM work? I think the quantization results of a single activation tile are reused by multiple GEMM tiles. Does this fusion lead to duplicated computation and lower the performance?

---

### Note · Authors · 2025-11-14

**Comment:**

We sincerely thank all reviewers for your valuable and constructive feedback. Upon careful reflection, we realize that the current experimental results in our submission are not sufficient to fully support our conclusions. Therefore, we have decided to withdraw the paper.

Since the initial submission, we have already conducted additional experiments, and we plan to further strengthen our analysis. We hope to present a more complete and well-supported version in our future submission.

**Withdrawal Confirmation:**

I have read and agree with the venue's withdrawal policy on behalf of myself and my co-authors.